# Combining Time-Restricted Wheel Running and Feeding During the Light Phase Increases Running Intensity Under High-Fat Diet Conditions Without Altering the Total Amount of Daily Running

**DOI:** 10.3390/ijms26157658

**Published:** 2025-08-07

**Authors:** Ayano Shiba, Roberta Tandari, Ewout Foppen, Chun-Xia Yi, Joram D. Mul, Dirk Jan Stenvers, Andries Kalsbeek

**Affiliations:** 1Netherlands Institute for Neuroscience, Institute of the Royal Netherlands Academy of Arts and Sciences (KNAW), Meibergdreef 47, 1105 BA Amsterdam, The Netherlands; a.shiba@nin.knaw.nl (A.S.); r.tandari@amsterdamumc.nl (R.T.); e.foppen@nin.knaw.nl (E.F.); c.yi@amsterdamumc.nl (C.-X.Y.); 2Laboratory of Endocrinology, Department of Laboratory Medicine, Amsterdam UMC, University of Amsterdam, Meibergdreef 9, 1105 AZ Amsterdam, The Netherlands; d.j.stenvers@amsterdamumc.nl; 3Amsterdam Gastroenterology, Endocrinology and Metabolism, 1105 AZ Amsterdam, The Netherlands; 4Department of Endocrinology and Metabolism, Amsterdam UMC, University of Amsterdam, Meibergdreef 9, 1105 AZ Amsterdam, The Netherlands; 5Brain Plasticity Group, Swammerdam Institute for Life Sciences, Faculty of Science, Science Park 904, 1098 XH Amsterdam, The Netherlands; j.d.mul@uva.nl; 6Centre for Urban Mental Health, University of Amsterdam, 1098 XH Amsterdam, The Netherlands

**Keywords:** circadian misalignment, time restricted running, time restricted feeding, liver, muscle, high fat diet, plin5

## Abstract

Excess caloric intake and insufficient physical activity are the two major drivers underlying the global obesity and type 2 diabetes mellitus epidemics. However, circadian misalignment of caloric intake and physical activity, as commonly experienced by nightshift workers, can also have detrimental effects on body weight and glucose homeostasis. We have previously reported that combined restriction of eating and voluntary wheel running to the inactive phase (i.e., a rat model for circadian misalignment) shifted liver and muscle clock rhythms by ~12 h and prevented the reduction in the amplitude of the muscle clock oscillation otherwise induced by light-phase feeding. Here, we extended on these findings and investigated how a high-fat diet (HFD) affects body composition and liver and muscle clock gene rhythms in male Wistar rats while restricting both eating and exercise to either the inactive or active phase. To do this, we used four experimental conditions: sedentary controls with no wheel access on a non-obesogenic diet (NR), sedentary controls with no wheel access on an HFD (NR-H), and two experimental groups on an HFD with simultaneous access to a running wheel and HFD time-restricted to either the light phase (light-run-light-fed + HFD, LRLF-H) or the dark phase (dark-run-dark-fed + HFD. DRDF-H). Consumption of an HFD did not alter the daily running distance of the time-restricted groups but did increase the running intensity in the LRLF-H group compared to a previously published LRLF chow fed group. However, no such increase was observed for the DRDF-H group. LRLF-H ameliorated light phase-induced disturbances in the soleus clock more effectively than under chow conditions and had a protective effect against HFD-induced changes in liver clock gene expression. Together with (our) previously published results, these data suggest that eating healthy and being active at the wrong time of the day can be as detrimental as eating unhealthy and being active at the right time of the day.

## 1. Introduction

Physical activity and food intake are closely intertwined, and the correct timing of these behaviours plays a significant role in the regulation of energy metabolism and homeostasis. When the timing of food intake and physical activity is not in line with the day–night cycle, which occurs during shift work, insomnia, or jet lag, it can lead to a situation in which peripheral and/or central clocks are no longer synchronized, also known as circadian misalignment. Circadian misalignment is known to impair glucose tolerance in shift workers, which can contribute to the development of type 2 diabetes mellitus [1]. In mouse models, which are nocturnal animals, food intake during their innate inactive phase (i.e., the light phase) also causes insulin resistance and exacerbates weight gain [2,3].

Current-day society is characterized by a dietary environment that is ‘unnaturally’ energy-dense and hyperpalatable, comes in virtually unlimited quantities, exploits the limbic system, and supersedes peripherally derived satiation and adiposity signals [4]. The well-described deleterious effects of a high-fat diet (HFD) consumption on the development of obesity can be further amplified by the erroneous timing of caloric consumption. Overconsumption of an HFD not only increases body weight, adiposity, glucose, and insulin levels [3] but also modifies the daily profile of wheel running behaviour and circadian synchronization to light [5]. Previously, we demonstrated in male Wistar rats that consumption of a free-choice high-fat high-sugar diet restricted to their innate inactive phase (i.e., the light phase) advanced hepatic clock rhythms [6] but did not reduce the amplitude of some soleus clock rhythms as observed with a chow diet restricted to the light phase [7]. Thus, both timing and nutritional factors seem to play a significant role in regulating clock gene expression rhythms of core metabolic organs (i.e., the liver and muscle).

Voluntary wheel running (VWR) has beneficial effects on several physical parameters related to metabolic health, such as body weight, fat content, and lean mass, and these beneficial effects can be further enhanced by a correct timing of physical activity [8,9,10,11]. In mice, VWR advanced the phase of the hepatic clock by 4 h in both chow- and HFD-fed conditions [12]. In rats, VWR in the inactive light phase did not shift the hepatic clock but did reduce amplitude of the soleus clock rhythms, whereas simultaneous time-restricted access to both a running wheel and a standard chow diet during the light phase successfully prevented the dampening of the soleus clocks and shifted both the hepatic and soleus clock rhythms by ~12 h, thus keeping them aligned [13].

Here, we investigated whether this combination of time-restricted voluntary wheel running and time-restricted feeding could ameliorate the negative metabolic health effects of a high-fat diet, such as increased body weight and fat gain, as well as prevent HFD-induced dysregulation of clock gene expression rhythms in liver and muscle.

## 2. Results

### 2.1. Effects of Simultaneous Voluntary Running and HFD Consumption Restricted to Dark or Light Phase on Locomotor Activity and Physiology

During the basal period of *ad libitum* running, DRDF-H and LRLF-H rats showed similar 24 h running patterns, with the majority of running performed during the dark (active) phase and similar gradual increases in their daily running distances (Figure 1A,G). Following the initiation of running restricted to either the active or inactive phase, DRDF-H animals ran daily for ~6 km per cage on average (i.e., ~3 km/day per rat), and LRLF-H animals ran an average of ~3 km daily per cage (i.e., ~1.5 km/day per rat) (Figure 1A). During this time restriction (TR) period, the LRLF-H cohort showed a strong running peak at the beginning of the light phase (ZT1-ZT5) followed by a sharp decline in running, whereas the DRDF-H group showed a more gradual increase in running activity, reaching a peak at ZT16-ZT17 followed by a gradual decrease (Figure 1H). During TR period, NR-H, DRDF-H, and LRLF-H groups consumed significantly more calories than the NR control group (Figure 1F). At the end of the experiment, DRDF-H rats displayed the lowest body weight and weight gain (Figure 1B,C). The body weight of the LRLF-H group was similar to that of the NR chow group, but weight gain during the TR period was significantly higher in the LRLF-H group than in the NR chow group and almost similar to that of the NR-H group (Figure 1B,C, Appendix A). Weight gain was highest in the NR-H and LRLF-H groups, which significantly differed from the DRDF-H group (Figure 1B,C, Appendix A). Both fat percentage and fat gain during TR were lowest in DRDF-H and highest in NR-H, while LRLF-H was in between (Figure 1D,E, Appendix A). At the end of the experiment, the fat percentage of the NR chow group was ~81% higher than that of the LRLF-H group, but fat gain of the NR group during the TR period did not differ from either DRDF-H or LRLF-H. This contrasts with the LRLF-H group that showed a significantly higher fat percentage and fat gain than the DRDF-H group (Figure 1D,E, Appendix A).

### 2.2. Comparing an HFD Versus Chow Diet During Time-Restricted Running and Feeding

Next, we compared the results from our current experiment with a previously published dataset in pair-housed male Wistar rats [13], in which both consumption of the non-obesogenic chow diet and voluntary running were restricted to either the active (dark) phase (DRDF) or inactive (light) phase (LRLF). Daily running distances were similar between DRDF and DRDF-H, and between LRLF and LRLF-H, both during the basal- and TR phases (Appendix A). Absolute body weight did not differ between the HFD and chow groups within the same restriction phase; however, a difference between the light and dark restricted groups was still present (Appendix A). The HFD resulted in a higher body weight gain during the TR period in LRLF-H compared to LRLF, but the weight gain of DRDF-H and DRDF did not differ (Appendix A).

In contrast to the body weight and body weight gain, fat percentage and fat gain did not differ between LRLF-H and LRLF, but fat gain was higher in DRDF-H than DRDF (Appendix A). Caloric intake was 30–36% higher in both HFD groups than in the chow groups (Appendix A). The daily running pattern of all groups were quite similar during the basal phase, although the HFD groups displayed a higher peak of running, which resulted in reaching its peak 1 h later than that of the chow groups (Appendix A). During the TR period, the two dark-restricted groups ran in a similar manner, with slightly higher levels in the chow group at the end of the dark period (Appendix A). The largest difference was observed between the two light-restricted groups: the LRLF-H group displayed only one sharp peak in the first half of the light phase, while the LRLF chow group displayed a first, smaller peak in the early light phase, followed by a second, even smaller peak later in the light phase (Appendix A). There was no significant difference in the total daily running distance between the chow and HFD groups during the time restriction (Appendix A).

### 2.3. Effect of Time-Restricted Running and Eating HFD on the Liver Clock

To assess how time restriction of feeding and running activities affects the liver clock under high-fat diet conditions, qPCR was performed on liver tissues collected across four different time points throughout the 24 h light/dark cycle.

In the liver, a significant *Group * Time* (interaction effect) was observed for all clock genes that were tested, except for *Per1* (Figure 2A, Appendix A). Cosinor analysis was performed to identify the acrophase (i.e., the time corresponding to the peak of the fitted gene oscillation curve), amplitude, and 95% confidence interval for all rhythms (Figure 2B, Appendix A). Several clock genes, including *Bmal1*, *Clock*, and *Reverba*, showed a trend towards a delayed phase in DRDF-H compared to NR-H, but a significant delay was only confirmed for *Bmal1* (3 h 47 m) (Figure 2B, Appendix A). All clock genes in LRLF-H showed a strong phase change in the acrophase compared with DRDF-H and NR-H. The magnitude of the phase difference varied among the genes. While LRLF-H was on average 8.96 ± 0.26 hrs ahead of DRDF-H in *Bmal1* and *Clock*, LRLF-H was on average more than 12 h (13.63 ± 1.58 h) ahead of DRDF-H in the rest of the clock genes (Appendix A). The remaining genes in the liver showed no significant effects of light- or dark-phase time-restricted running combined with feeding on amplitude.

### 2.4. Effect of Time-Restricted Running and Eating HFD on the Soleus Clock

In the soleus muscle, a significant *Group * Time* (interaction effect) was observed for all clock genes that were tested (Figure 3A, Appendix A). Cosinor analysis revealed that all clock genes were shifted by LRLF-H but with various magnitudes, like in the liver (Figure 3B, Appendix A). Although phases of DRDF-H and LRLF-H were about 9+ h apart in the liver clock, the phase difference was as low as 6.6 h in several soleus clock genes. This seems to be caused mainly by the phase changes in DRDF-H. DRDF-H advanced the phases of *Bmal1* and *Per2* by on average 4.59 ± 0.79 h and delayed the phase of *Cry1* by 3.8 h compared to NR-H. The amplitude of Per2 was reduced by LRLF-H compared with DRDF-H and NR-H. The amplitudes of the remaining genes were not affected by light- or dark-phase running plus feeding.

### 2.5. Effect of Time-Restricted HFD and Running on the Lipid Droplet Associated Protein Gene

Recently, [14] found that time-restricted feeding during the light phase increases running endurance and induces a daily rhythm of muscle *Plin5*. Therefore, we investigated the gene expression of the lipid droplet-associated protein perilipin-5 (*Plin5*) in the soleus muscle. Significant *Group* and *Group * Time* (interaction effect) effects were observed for Plin5 when on HFD, while only a *Group* effect was found when on chow (Figure 4A, Appendix A). Cosinor analysis revealed no amplitude changes triggered by time restriction in muscle *Plin5* expression under either chow or HFD conditions (Figure 4A,C, Appendix A). The mean *Plin5* expression levels were significantly increased by HFD in NR-H versus NR mice (Figure 4B bottom). At the same time, the time-restricted running and feeding combination reduced mean *Plin5* expression in both chow and HFD conditions (Figure 4B top and bottom).

In the chow condition, the *Plin5* acrophase was advanced by ~5 h by LRLF compared with NR (Figure 4C top). In the HFD condition, the phase of DRDF-H *Plin5* was delayed by 4.3 h compared to that of NR-H (Figure 4C bottom, Appendix A). LRLF-H *Plin5* was phase-delayed by 14.5 h compared with NR-H, and 7.4 h compared with DRDF-H (Figure 4C bottom).

## 3. Discussion

Here, we report that time-restricted feeding in combination with time-restricted running is not only beneficial during consumption of a chow diet (as published before [13]), but also when animals are eating a high-fat diet (HFD). The dark-phase running and feeding group on a high-fat diet (DRDF-H) had a lower body weight and fat mass percentage not only compared to the sedentary group on a high-fat diet (NR-H) and the light-phase running and feeding group on a high-fat diet (LRLF-H), but even compared to the sedentary group on a chow diet (NR). Remarkably, the LRLF-H group displayed a larger initial running peak than the LRLF group on chow, although the total daily running distance during the time restriction period did not differ (Appendix A). No such differences in running activity were observed between the DRDF-H and DRDF groups. The hepatic clock gene expression patterns of the time-restricted HFD groups were similar to those of our previous time-restricted cohorts on a chow diet [13], suggesting that the combination of LR and LF prevented the reduction in the clock gene oscillation amplitude in hepatic clock genes caused by light-phase feeding (LF) in combination with HFD, as observed in our previous study [6] (Appendix A). In contrast to our previous chow time-restricted feeding and running study that was performed with the exact same experimental setup and design as the current study [13], the acrophase and amplitude of some soleus clock genes were affected by the time-restricted protocol under HFD conditions (Appendix A). The mean relative expression of the lipid droplet-associated *Plin5* gene was increased by HFD and decreased by the combination of time-restricted feeding and running in both chow and HFD rats. The expression rhythm of *Plin5 was* shifted in the light phase time-restricted feeding and running groups under both chow and HFD conditions (See Figure 5 for an overview of our findings).

While the weight gain of LRLF-H was higher than that of chow NR, the fat gain of LRLF-H did not differ from that of NR, which indicates an increased fat-free mass (e.g., muscle) gain in LRLF-H compared to NR. The time-restricted feeding and exercise groups on a high-fat diet (DRDF-H and LRLF-H) showed no difference in caloric intake compared to the sedentary high-fat diet group (NR-H). This contrasts to our previous chow results [13], in which the dark phase restricted (DRDF) group consumed significantly more calories than the NR group, while the chow light phase-restricted (LRLF) group consumed fewer calories than the DRDF group. In our previous time-restricted feeding study using a free-choice high-fat high-sucrose (fcHFHS) diet, we found that time-restricted feeding reduced the caloric intake of chow, but did not affect total fat and sucrose intake compared to *ad libitum* feeding [6]. This preference for high caloric food potentially explains the absence of an effect of time restriction on total caloric intake in the current HFD study in contrast to our previous chow study [13].

When comparing the current HFD study with our previous chow study, the DRDF-H and DRDF groups showed similar body weight increase, but fat gain was still significantly higher in DRDF-H than in DRDF. Therefore, DRDF could not completely prevent the effects of HFD. In other words, the correct timing of food intake and physical activity may partly, but not fully, reduce the negative metabolic effects of HFD. On the other hand, fat mass and gain of fat mass were similar between the LRLF and DRDF-H groups, indicating that eating healthy but being active at the wrong time of the day is as detrimental as eating unhealthy and being active at the right time of the day. This is in line with human studies reporting that exercise during the late active phase (at 18:30, maximum of 1 h) only partly reversed HFD-induced changes in the metabolic profile [15]. Studies in male Swiss mice fed an HFD also reported that only time-restricted feeding during the dark phase (TRF, ZT 16–0) combined with exercise (ZT0–1), but not time-restricted feeding alone, improved respiratory exchange rate, energy expenditure, mitochondrial respiration, decreased lipogenic and glucogenic gene expression, and attenuated HFD-induced changes in the insulin signaling pathway in the liver [16,17]. In addition, voluntary wheel running did not fully counteract the negative metabolic effects of HFD, including adiposity and free cholesterol levels in mice [18]. Thus, both time-restricted feeding and exercise are effective methods to combat the negative effects of an HFD but may not be able to completely compensate for these effects depending on the metabolic measurement of interest.

In our previous study, the soleus *Per1* and *Reverb-a* rhythms were dampened by light running (LR), and the combination of LR and LF partially prevented this dampening [13]. The current study shows that under HFD conditions, the LRLF combination successfully prevented this dampening effect of eating at the wrong time of day. However, this “rescuing” effect of LRLF does not work for all genes, as the soleus *Per2* rhythm was still dampened in LRLF-H. Considering studies reporting that both LF and HFD reduces soleus *Per2* expression [19], and both HFD-dark-phase restricted feeding and HFD-*ad libitum* feeding prevent dampening of soleus *Per2* [7], soleus *Per2* is likely strongly regulated by both dietary content and timing. As *Per2* directly regulates PPARγ activity, which is a nuclear receptor essential for lipid metabolism, based on the adiposity and other metabolic outcomes, it is not surprising that the “rescuing” effect of wheel running was limited when performed at the wrong time of the day (i.e., LRLF and LRLF-H) [20].

In the liver too, the results of the LRLF-H group revealed a protective effect of the LRLF combination on the dampening of some core clock gene rhythms triggered by HFD. A previous study showed that HFD in combination with LF either dampened (*Per2* and *Reverba*) or caused a small phase advance (*Bmal1* and *Cry1*) in core clock gene expression rhythms in the liver [6]. However, we found no dampening or small phase shifts in LRLF-H compared to the LRLF chow group, suggesting that the combination of LR and LF has a stronger influence on liver clock gene rhythms than HFD. This is in line with the observation that TRF prevented a loss of diurnal rhythmicity that occurred during HFD-induced obesity in mice [21], also suggesting that the influence of TRF on diurnal rhythmicity is more powerful than that of HFD consumption.

In our previous study with a chow diet, the acrophase of DRDF was similar to that of NR and was also in antiphase with LRLF in both the liver and the soleus muscle. In this study, a phase advance of DRDF-H compared to NR-H was observed in three out of the seven soleus clock genes tested. Moreover, the remaining four genes showed a trend toward a phase advance. As a result, the majority of clock genes no longer showed a clear antiphase between LRLF-H and DRDF-H. In the liver, a phase advance of DRDF-H compared to NR-H was only observed for *Bmal1*, while most other genes retained the antiphase between DRDF-H and LRLF-H despite the HFD.

The current study demonstrated the gene specific response of clock genes towards HFD in combination with time-restricted running and feeding. An increasing number of studies have provided evidence that clock gene responses are dependent on organs, tissue type, feeding timing, diet, exercise timing, the combination of both, and many other factors [22,23,24]. One study reported that TRF alone can affect the expression or rhythmicity of 80% of all genes in at least one tissue [25]. Further studies on clock gene expression rhythms and their specificity are needed to provide the necessary insights on how to prevent or repair circadian misalignment to treat or prevent its negative health effects.

A previous study has reported that LF enhances running endurance via myocyte *Plin5* and induces a daily rhythm of muscle *Plin5* expression in a clock-dependent manner in mice [14]. In the current study, we only found a higher activity peak in the LRLF-H versus LRLF group at the beginning of the light period, which may be linked to increased running endurance. No differences in total amount of running were found between LRLF-H and DRDF-H, although running in the LRLF-H animals was concentrated in 4 h versus 10 h in the DRDF-H group. In addition to the mouse versus rat difference, the major difference is that while Xin et al. (2023) utilised a treadmill and a forced running model to assess maximal running capacity, we used a voluntary running model [14,26]. Although several studies have shown that voluntary running increases running endurance [27,28,29,30], our observations may not strictly increase endurance, but rather increase running intensity, as it does not challenge the maximal exercise capacity of rats.

Several studies agree that in male Wistar rats, running at 22–25 m/min without a slope is considered high-intensity, and 22 m/min for 50–60 min is commonly used for an endurance treadmill training model that corresponds to a maximal oxygen consumption (VO2max) range of 65–75% [31,32,33,34,35]. In the present study, the LRLF-H group max. ran at 450 m/h on the running wheel, which is equivalent to 7.5 m/min. Running distances differ depending on the type of running wheel, for instance, the average running distance for an angled wheel is 10–20 km/day, while for an upright running wheel, it is 8–10 km/day in mice [29]. Taken together, the intensity of voluntary running in the LRLF-H group was equivalent to more than 15 m/min on a treadmill, which is considered moderate exercise. However, differences in size, material, voluntary vs. forced running, and several other factors make it challenging to compare the exercise load between a treadmill and a running wheel.

Previous studies have also reported that HFD upregulates *Plin5* expression in the heart [36], liver [37,38], and muscle tissues [39,40]. In this study, we confirmed a significant increase in muscle *Plin5* expression in HFD NR compared to chow NR. In addition, we found that under both chow and HFD conditions, relative *Plin5* expression was downregulated by the combination of time-restricted running and feeding compared to the non-runners. These results are in line with previous studies in both mice and humans. A study in mice demonstrated that strength training decreases *Plin5* level in heart and skeletal muscle tissues [41]. Also, human muscle biopsy results from Xin et al. (2023) confirmed that exercise reduces Plin5 expression in a time-of-day-dependent manner in both young and adult cohorts [14].

Despite the increased wheel running intensity in the early light phase in the LRLF group on an HFD, we did not find the time-of-day-dependent increase in muscle *Plin5* expression reported in rats. To our knowledge, there have been no studies on the combined effect of LF and exercise on *Plin5* expression, where the stimulatory effect of LF on *Plin5* expression may be antagonized by the inhibitory effect of exercise. Based on the result of the current study, it is likely that the effect of LR is stronger than that of LF. However, another possibility is the experimental duration. Plin5 in myokines has been reported to drastically increase FGF21 expression in muscle and elevate serum FGF21 concentration [42]. However, this effect seems to be dependent on the experimental duration. While acute exercise increases circulatory FGF21 levels, chronic exercise lasting four or more weeks has the opposite effect [43]. Under HFD conditions, it has been demonstrated that wheel running causes no change in *Plin5* mRNA nor protein expression [37], which is different from our current results. This discrepancy could be due to a difference in the composition of the high-fat diet or species differences. Lack of intensity or duration of the wheel-running activity is unlikely to be the contributing factor. The duration of our experiment was about seven weeks, while the experiment of Tuikka et al. lasted ten weeks. Regarding running distance, in the study of Tuikka et al. the maximum running distance between chow and HFD did not differ in the first part of the experiment, while the general running distance was double in chow than those of HFD in the 2nd half of the experiment [37]. This difference is similar to that observed between running in the active-phase and inactive-phase [13]. The fact that the decrease in Plin5 in our study was observed in both LRLF and DRDF groups under both chow and HFD conditions suggests that the running distance in the experiment of Rinnankoski-Tuikka was likely sufficient.

## 4. Materials and Methods

### 4.1. Animals and Housing

A total of 80 male Wistar rats (WU, Charles River, Sulzfeld, Germany) were used for this study, weighing 285 (±23) grams (8–9 weeks old) upon arrival in two batches of 40 rats. After 1 week of acclimatization, rats were pair-housed in a temperature- (21–23 °C), humidity- (40–60%), and light-controlled room [12:12 h light/dark cycle; 280 (±80) Lux (light phase): <5 Lux (dark phase); with lights on at Zeitgeber time (ZT) 0 (i.e., the start of innate inactive phase) and lights off at ZT12 (i.e., the start of innate active phase)] in the animal facility of the Netherlands Institute for Neuroscience. ZT0 was 08:00 during the summertime, and 07:00 during the wintertime (daylight saving time). Rats were housed in custom-made cages [522 (w) × 582 (l) × 412 (h) mm], in which they could freely move between the home cage compartment and a vertical 36 cm-diameter stainless-steel running wheel (Model 80850MS, Campden Instruments, Loughborough, UK). An irradiated nutritionally complete high-carbohydrate pelleted chow diet (#2918, Teklad Irradiated Global 18% Protein Rodent Diet, 24% kcal from protein, 58% kcal from carbohydrate, and 18% kcal from fat, 3.1 kcal/g, Inotiv, Lafayette, IN, USA) or an obesogenic HFD (#3282, Mouse Diet High Fat Calories 60%, 15% kcal from protein, 26% kcal from carbohydrate, and 59% kcal from fat, 5.49 kcal/g, Bio Serv, Plexx B.V., Elst, The Nehterlands) and a bottle with tap water were available *ad libitum* in the home cage throughout the experiment. Further details of the running wheel setup have been described previously [13].

### 4.2. Experimental Design

#### 4.2.1. Experimental Cohorts for RT-qPCR Analysis

After 10 days of acclimatization, rats were randomly assigned to one of three groups: sedentary chow (NR; no access to running wheel) (*n* = 8), sedentary high fat (NR-H; no access to running wheel) (*n* = 24), and access to the running wheel with HFD (*n* = 48). Body weight was measured weekly. Fat mass was also measured weekly using an EchoMRI-500 QMR system (EchoMRI, Houston, TX, USA). Food intake was measured 2–3 times weekly to estimate daily food intake. During the basal period (days 0–18), all rats with a running wheel had *ad libitum* access to become accustomed to them. After the basal period, animals with a running wheel were randomly divided into either dark running plus dark-fed HFD (Dark-Run-Dark-Fed; DRDF-H; *n* = 24) or light running plus light-fed HFD (Light-Run-Light-Fed; LRLF-H; *n* = 24) groups based on body weight, fat percentage, and running activity using RandoMice (LUMC, Leiden, The Netherlands) [44]. During the 28 days of time restriction (day 18–46), access to the wheels and food was restricted to a 10 h time period in the dark (ZT13-23 for the DRDF-H group) or light period (ZT1-11 for the LRLF-H group). The NR and NR-H control groups were housed without running wheels throughout the experiment and with *ad libitum* food access. Animals in the NR group were only used for the measurements of body weight and fat percentage but excluded from RT-qPCR experiments due to limited number of animals per time point. At the end of the time restriction, rats on the HFD were sacrificed by CO_2_ sedation followed by decapitation at ZT0, ZT6, ZT12, or ZT18, and the liver (left lateral lobe) and right soleus muscles were rapidly dissected, snap-frozen in liquid nitrogen, and stored at −80 °C until further analysis. Sample sizes were *n* = 6 for all three groups at each of the four time points. In order to minimize the number of animals used for the current study according the 3R principle, we did not include no-running (sedentary) controls with time-restricted feeding (LF, DR) and reduced the number of NR animals. Consequence of the reduced number of NR animals was that we could not include them in the current qPCR experiment. However, daily clock gene expression patterns of these three control groups have been reported previously in studies using the exact same experimental setup and design [6,7,13].

#### 4.2.2. RNA Isolation, cDNA Synthesis, and RT-qPCR

RNA isolation, cDNA synthesis, and RT-qPCR were performed as described earlier [7]. All the primer sequences and housekeeping genes used for each tissue are listed in Appendix A.

### 4.3. Statistics

GraphPad Prism8 (CA, USA) was used to perform statistical tests and visualize the data. Normal distribution was confirmed using a normal QQ plot. Assessment of effects in experiments involving several conditions was performed using one-way analysis of variance (ANOVA) or mixed-effects analysis, with repeated measures where applicable, followed by Tukey’s HSD post hoc tests to adjust for multiple comparisons when appropriate. Data are presented as mean ± SEM. RT-qPCR data were analyzed using LightCycler (Roche, Basel, Switzerland) conversion and LinReg software version 2021.1 (AMC, Amsterdam, The Netherlands). Cosinor-based rhythmometry analyses were performed with CosinorPy (Ljubljana, Slovenia) [45] using Python 3.8.5. (Anaconda, TX, USA).

## 5. Conclusions

We found that the combination of time-restricted feeding and time-restricted running ameliorates high-fat diet-induced weight gain and body fat accumulation, ameliorates LF-induced disturbances in the soleus clock more effectively than under chow conditions, and has a protective effect against high-fat diet-induced changes in liver clock gene expression. Taken together, the combination of time-restricted feeding and time-restricted running helps to partially prevent the negative effects of an HFD on both metabolic health and clock gene expression rhythms.

## Figures and Tables

**Figure 1 ijms-26-07658-f001:**
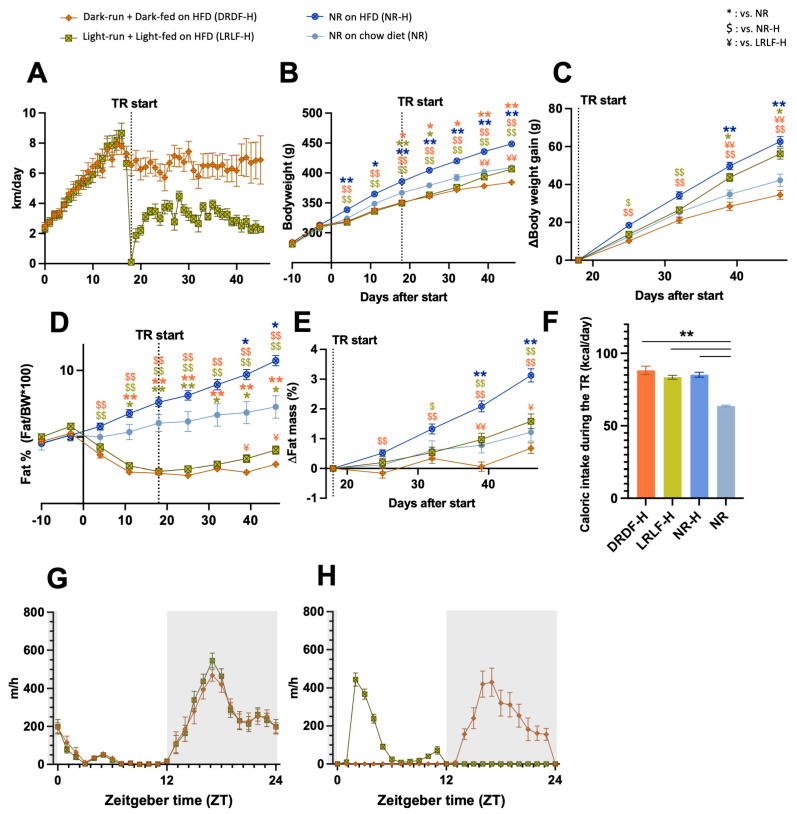
Under high-fat diet (HFD) conditions combining time-restricted wheel running and time-restricted feeding in the light phase has less beneficial metabolic outcomes than combining running and feeding in the dark phase. (**A**): The average daily running distance of each group per animal during the experiment. (**B**): Body weight growth. (**C**): Gain of body weight from the start of the time restriction. (**D**): Fat mass in percentage of body weight. (**E**): Gain of body fat from the start of the time-restricted running period. (**F**): Caloric intake during the time restriction period. (**G**,**H**): Daily running pattern during the baseline (**G**) and time restriction (**H**). Dark running dark-fed group on high fat diet (DRDF-H, in orange with brown frame): *n* = 24, Light running light-fed group on high fat diet (LRLF-H, in yellow green with olive frame): *n* = 24, Sedentary non-runner on high fat diet (NR-H, blue with dark blue frame): *n* = 24, Sedentary non-runner on chow diet (NR, light blue with blue frame): *n* = 8. Data are presented as the mean ± SEM. Significant difference from NR (*), from NR-H ($), or from LRLF-H (¥) compared to the groups of color code. * or $ or ¥: *p* < 0.05, ** or $$ or ¥¥: *p* < 0.01 by one-way ANOVA or mixed-effects analysis followed by Tukey’s HSD post hoc test. TR: time restriction.

**Figure 2 ijms-26-07658-f002:**
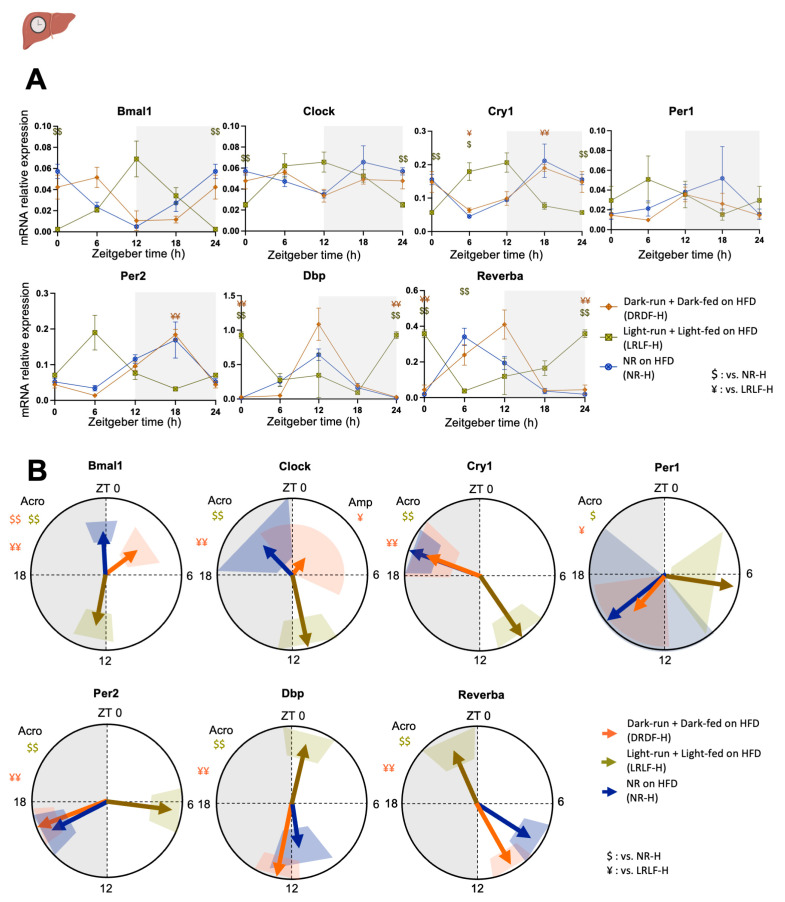
Four weeks of combined running and feeding during the light phase alters the expression profiles of clock genes in rat liver. (**A**): mRNA relative expression analyzed by two-way ANOVA followed by Tukey HSD post hoc test. (**B**): Acrophase (indicated by the direction of arrows) with their amplitude (indicated by the length of arrows) of the clock (controlled) genes were analysed by cosinor-based rhythmometry analysis using CosinorPy. CosinorPy adjusts the significance values using the false discovery rate (FDR) method (reported as *q*-values). Signs in the right top corner of each circular figure represent significant differences in amplitude. Signs in the left top corner of each circular figure represent significant differences in acrophase. Grey shaded area represents the dark (inactive) phase. Coloured shaded areas (corresponding to the group colour code) in B represent 95% confidence interval. ZT = Zeitgeber time, h = hour (time). Amp: Amplitude. Acro: Acrophase. Data are presented as the mean ± SEM. Significant difference from NR-H ($), or from LRLF-H (¥) compared to the groups of color code. $ or ¥: *p* < 0.05, $$ or ¥¥: *p* < 0.01 by one-way ANOVA or mixed-effects analysis followed by Tukey HSD post hoc test. Icon created with BioRender.com.

**Figure 3 ijms-26-07658-f003:**
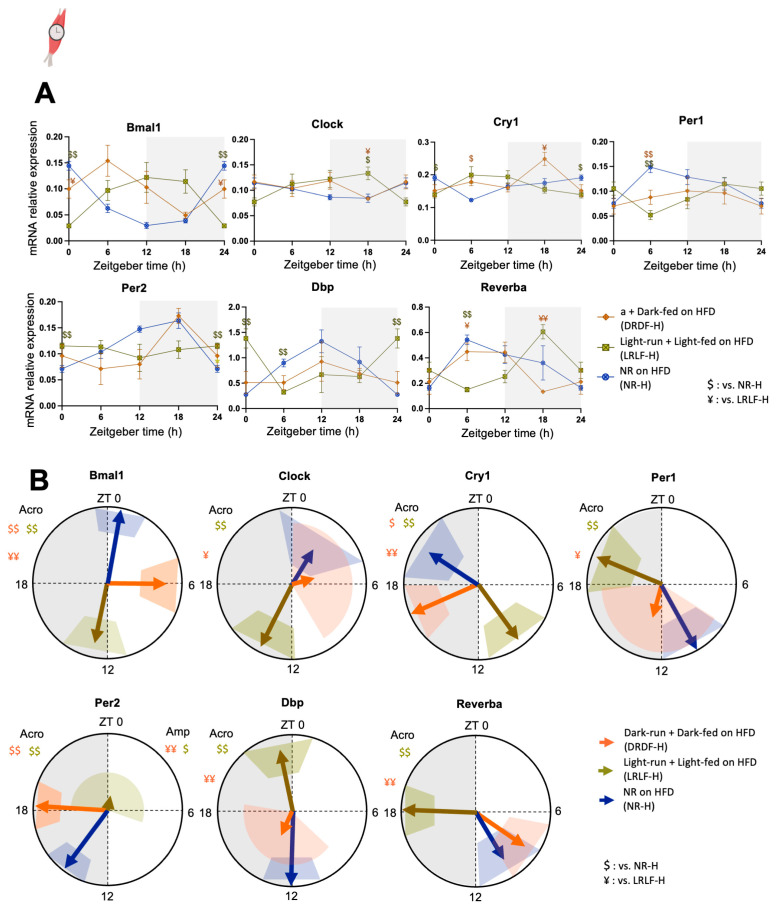
Four weeks of combined running and feeding during the light phase alters the expression profiles of clock genes in rat soleus. (**A**): mRNA relative expression analyzed by two-way ANOVA followed by Tukey HSD post hoc test. (**B**): Acrophase (indicated by the direction of arrows) with their amplitude (indicated by the length orestriction period did not differ (Appendix A). No such differences in running activity were observed between the DRDF-H and DRDF groups. The hepatic clock gene expression patterns of the time-restricted HFD groups were similar to those of our previous time-f arrows) of the clock (controlled) genes were analysed by cosinor-based rhythmometry analysis using CosinorPy. CosinorPy adjusts the significance values using the false discovery rate (FDR) method (reported as *q*-values). Signs in the right top corner of each circular figure represent significant differences in amplitude. Signs in the left top corner of each circular figure represent significant differences in acrophase. Grey shaded area represents the dark (inactive) phase. Coloured shaded areas (corresponding to the group colour code) in B represent 95% confidence interval. ZT = Zeitgeber time, h = hour (time). Amp: Amplitude. Acro: Acrophase. Data are presented as the mean ± SEM. Significant difference from NR-H ($), or from LRLF-H (¥) compared to the groups of color code. $ or ¥: *p* < 0.05, $$ or ¥¥: *p* < 0.01 by one-way ANOVA or mixed-effects analysis followed by Tukey’s HSD post hoc test. Icon created with BioRender.com.

**Figure 4 ijms-26-07658-f004:**
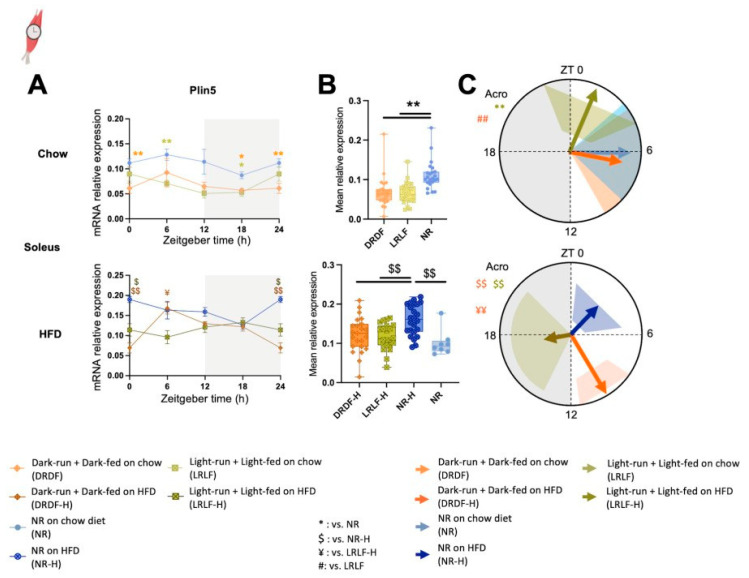
Both high-fat diet (HFD) and four weeks of combined running and feeding alters the expression levels and profiles of *Plin5* gene in rat soleus muscle. (**A**): mRNA relative expression analyzed by two-way ANOVA followed by Tukey HSD post hoc test. (**B**): Mean *Plin5* expression in soleus muscle under both chow (top) and high fat diet condition (HFD, bottom). (**C**): Acrophase (indicated by the direction of arrows) with their amplitude (indicated by the length of arrows) of the clock (controlled) genes were analysed by cosinor-based rhythmometry analysis using CosinorPy (top: chow, bottom: HFD). CosinorPy adjusts the significance values using the false discovery rate (FDR) method (reported as *q*-values). Signs in the right top corner of each circular figure represent significant differences in amplitude. Signs in the left top corner of each circular figure represent significant differences in acrophase. Grey shaded area represents the dark (inactive) phase. Coloured shaded areas (corresponding to the group colour code) in B represent 95% confidence interval. ZT= Zeitgeber time, h= hour (time). Amp: Amplitude. Acro: Acrophase. Chow; At ZT0 (NR, *n* = 6; DRDF, *n* = 6; LRLF, *n* = 6), at ZT6 (NR, *n* = 6; DRDF, *n* = 6; LRLF, *n* = 6), at ZT12 (NR, *n* = 6; DRDF, *n* = 6; LRLF, *n* = 6), and at ZT18 (NR, *n* = 6; DRDF, *n* = 6; LRLF, *n* = 6). Data are presented as the mean ± SEM. Significant difference from NR (*), from NR-H ($), or from LRLF-H (¥), or from LRLF, compared to the groups of color code. * or $ or ¥ or #: *p* < 0.05, ** or $$ or ¥¥ or ##: *p* < 0.01 by one-way ANOVA or mixed-effects analysis followed by Tukey’s HSD post hoc test. Icon created with BioRender.com.

**Figure 5 ijms-26-07658-f005:**
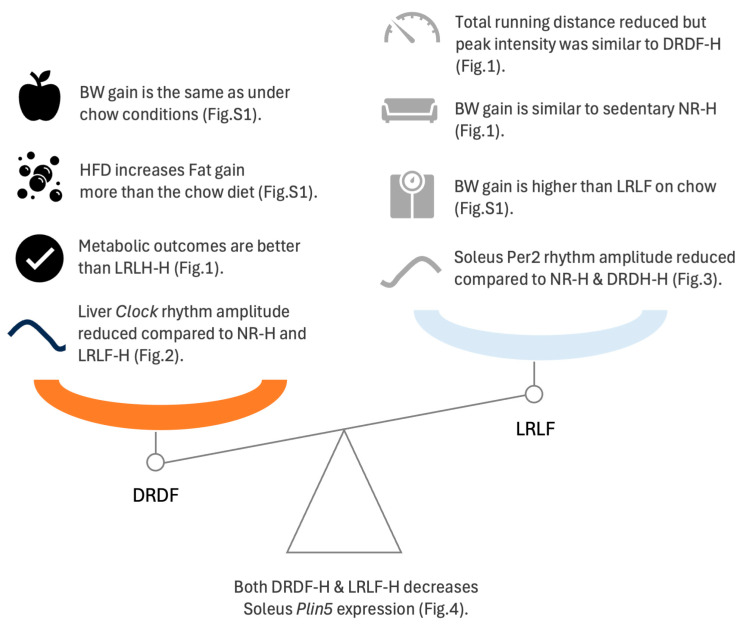
Summary of the current findings on the impact of HFD and the combination of time-restricted running and time-restricted feeding.

## Data Availability

Data will be made available on reasonable request.

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
