# Peer review of "Combining Time-Restricted Wheel Running and Feeding During the Light Phase Increases Running Intensity Under High-Fat Diet Conditions Without Altering the Total Amount of Daily Running"

_ijms, 2025, doi:10.3390/ijms26157658_

Round 1
Reviewer 1 Report
Comments and Suggestions for Authors
The manuscript titled “Combining time-restricted wheel running and feeding during the light phase increases running intensity under high-fat diet conditions without altering the total amount of daily running” by Shiba et al. described the effect of exercise (free running-wheel) and day/ night phase on rats under a high-fat diet. The Authors found out that the combination of feeding and allowing the rats to exercise influences the body weight. The manuscript is well-written, clear, and employs the correct statistical analyses.
There are minor comments regarding a few aspects of the manuscript, as detailed below.
Line 2: Please correct the typo by adding a hyphen on “high fat”.
Line 50: To facilitate the understanding of the experimental design, it is suggested to either simplify the abbreviations or define them based on their meaning. The definitions given on line 124 could also be introduced here for better understanding.
Line 171: The Authors should provide the F-values from the ANOVA for all results that involved ANOVA.
References: Despite the lower number of papers in the area, the Authors should include additional evidence from the literature to either support or contrast their results. Most of the discussion is based on their previous manuscript.
Author Response
For research article
Response to Reviewer 1 Comments
|
||
1. Summary |
|
|
Thank you very much for taking the time to review this manuscript. Please find the detailed responses below and the corresponding revisions/corrections highlighted/in track changes in the re-submitted files. Modifications on the manuscript in response to the comments are highlighted with light green marker.
|
||
2. Questions for General Evaluation |
Reviewer’s Evaluation |
Response and Revisions |
Does the introduction provide sufficient background and include all relevant references? |
Can be improved |
We have tried to improve the Introduction section, where possible, to increase clarity for the reader. |
Are all the cited references relevant to the research? |
Yes |
|
Is the research design appropriate? |
Yes |
|
Are the methods adequately described? |
Yes |
|
Are the results clearly presented? |
Can be improved |
We have tried to improve the Results section, where possible, to increase clarity for the reader. |
Are the conclusions supported by the results? |
Yes |
|
3. Point-by-point response to Comments and Suggestions for Authors |
||
Comments 1: Line 2: Please correct the typo by adding a hyphen on “high fat”. |
||
Response 1: Thank you. A hyphen has been added.
|
||
Comments 2: Line 50: To facilitate the understanding of the experimental design, it is suggested to either simplify the abbreviations or define them based on their meaning. The definitions given on line 124 could also be introduced here for better understanding. |
||
Response 2: We have added now an explanation for the abbreviations used.
Comments 3: Line 171: The Authors should provide the F-values from the ANOVA for all results that involved ANOVA. Response 3: F-values are now added to supplemental figures next to p-values.
Comments 4: References: Despite the lower number of papers in the area, the Authors should include additional evidence from the literature to either support or contrast their results. Most of the discussion is based on their previous manuscript. Response 4: We have added the following paragraphs and references:
Line 303-309: Considering studies reporting that both LF and HFD reduces soleus Per2 expression [21], and both HFD-dark-phase restricted feeding and HFD-ad-libitum feeding prevent dampening of soleus Per2 [7], soleus Per2 is likely strongly regulated by both dietary content and timing. As Per2 directly regulates PPARγ activity, which is a nuclear receptor essential for lipid metabolism, it is not surprising that the “rescuing” effect of wheel running when performed at the wrong time of the day (i.e. LRLF and LRLF-H) was limited based on adiposity and other metabolic outcomes [22].
Line 316-318: This is in line with the observation that TRF prevented a loss of diurnal rhythmicity that occurred during HFD-induced obesity in mice [24], suggesting that the influence of TRF on diurnal rhythmicity is more powerful than that of HFD consumption.
|

Reviewer 2 Report
Comments and Suggestions for Authors
The paper is very interesting and significant in the context of understanding the internal clocks in the body and their misalignment due to external factors such as feeding and physical activity or shifting sleep periods. I think the paper would benefit from adding a number of explanations as well as better specifying the use of data from another study. I suggest that the following corrections and additions be made:
- Please define (using precise timing) what you mean when you say inactive/active phase, dark/light period of the day. This is different for rats and humans. Although it should be the generally known time for lights on and off, in all animal housing facilities it is not the same. Please define light/dark/sedentary/inactive/active in introduction.
- When talking about the shift of the circadian clock, please use mathematical notation (+/-) in body text to make it clearer what kind of shift is occurring. The word dampening doesn't really explain what you meant.
- In your experiment you had two groups of animals on HFD that had time restricted feeding and running or in light or dark phase. In order to have proper controls for them, you need two groups of animals that are only on HFD, without physical activity and have time-restricted feeding in the light or dark phase. Alternatively, your control could have been animals on time restricted feeding (light/dark phase) with standard diet without physical activity. You can use animals from other experiments as controls, but you must provide evidence that they were kept under the same conditions, with the same food, the same running-wheels, and the same personnel who cared for them. You can justify this with the 3R principle. In order to do this, you also need a short description of that previous experiment (age, sex of animals, duration of the experiment) so that it is clear whether it is comparable or not without reading your previous paper.
- You also have two control groups of animals, one on a standard diet and the other on a HFD, and that's fine, but you should emphasize that the feeding of these animals was ad libidum and not time restricted, because you were only monitoring standard conditions and the shift in the circadian clock by simply changing the type of diet.
- The last sentence of the abstract would be an excellent conclusion if you had LRLF and DRDF on a standard diet in this trial or if you had LF-H and DF-H. You cannot draw this conclusion based on the description of the groups in the second paragraph of the abstract alone.
- Explain Zeitgeber time and acrophase to the readers, i.e. connect ZT0 with a point in standard time (morning point). The topic is interesting and will surely be read by those who are not familiar with all the technical terms.
- Explain why the animals in the NR and NR-H group were not used for RT-qPCR when they are the control for the standard circadian rhythm, i.e. the shift in it induced only by food.
- It is awkward to say body weight development (Fig 1B), body weight evolution (line 272)
- Carefully list the limitations of the study that your study has with respect to the controls you use from your other studies.
Author Response
For research article
Response to Reviewer 2 Comments
|
||
1. Summary |
|
|
Thank you very much for taking the time to review this manuscript. Please find the detailed responses below and the corresponding revisions/corrections highlighted/in track changes in the re-submitted files. Modifications that were made in response to reviewers comments on the manuscript are highlighted with light green marker.
|
||
2. Questions for General Evaluation |
Reviewer’s Evaluation |
Response and Revisions |
Does the introduction provide sufficient background and include all relevant references? |
Can be improved |
We have tried to improve the Introduction section, where possible, to increase clarity for the reader. |
Are all the cited references relevant to the research? |
Must be improved |
We have tried to improve the references throughout the manuscript, where possible, to increase clarity for the reader. |
Is the research design appropriate? |
Can be improved |
The research design cannot be changed in hindsight, but we have tried to explain the rationale of our design better. |
Are the methods adequately described? |
Yes |
|
Are the results clearly presented? |
Must be improved |
We have tried to improve the Results section, where possible, to increase clarity for the reader. |
Are the conclusions supported by the results? |
Yes |
|
3. Point-by-point response to Comments and Suggestions for Authors |
||
Comments 1: Please define (using precise timing) what you mean when you say inactive/active phase, dark/light period of the day. This is different for rats and humans. Although it should be the generally known time for lights on and off, in all animal housing facilities it is not the same. Please define light/dark/sedentary/inactive/active in introduction. |
||
Response 1: We agree with the reviewer that this terminology should be accurate. We hence made the following modifications to increase clarity for the reader: Line 71: We added “nocturnal animals“ after “mouse models”, and “innate” before inactive followed by “(i.e. the light phase)”. Line 82: We added “innate” before inactive followed by “(i.e. the light phase)”, Line 92: “inactive” followed by “light”. Line 94: “inactive” replaced with “light”. Line 109-111: ZT0 was 08:00 during the summer time, and 07:00 during the winter time (daylight saving time).
|
||
Comments 2: When talking about the shift of the circadian clock, please use mathematical notation (+/-) in body text to make it clearer what kind of shift is occurring. The word dampening doesn't really explain what you meant. |
||
Response 2: We agree with the reviewer that related sentences should be accurate. However, we think that “advancing” and “delaying” a phase shift is a better description than “a phase shift of +4 hours or -3 hours”. With “dampening”, we meant a reduced amplitude of the clock gene oscillation. To improve the clarity, we redefined the word dampening. We hence made the following modifications to increase clarity for the reader: Line 92: we replaced dampens for “reduced amplitude”. Line 258-259: we replaced dampening for “the reduction of the clock gene oscillation amplitude”.
Comments 3: In your experiment you had two groups of animals on HFD that had time restricted feeding and running or in light or dark phase. In order to have proper controls for them, you need two groups of animals that are only on HFD, without physical activity and have time-restricted feeding in the light or dark phase. Alternatively, your control could have been animals on time restricted feeding (light/dark phase) with standard diet without physical activity. You can use animals from other experiments as controls, but you must provide evidence that they were kept under the same conditions, with the same food, the same running-wheels, and the same personnel who cared for them. You can justify this with the 3R principle. In order to do this, you also need a short description of that previous experiment (age, sex of animals, duration of the experiment) so that it is clear whether it is comparable or not without reading your previous paper. Response 3: We agree with the reviewer that the inclusion of several appropriate control groups would have been optimal. However, as rightfully noted by the reviewer, this would have increased the number of experimental animals tremendously, which would not be in accordance with the 3R principles. Time-restricted feeding experiments comparing rats on a chow diet versus a HFD have been performed previously in our group, see for example J. E. Oosterman et al., “Synergistic Effect of Feeding Time and Diet on Hepatic Steatosis and Gene Expression in Male Wistar Rats,” Obesity, 2020 [ref 6 in ms] or P. de Goede et al., “Differential effects of diet composition and timing of feeding behavior on rat brown adipose tissue and skeletal muscle peripheral clocks,” Neurobiol. Sleep Circadian Rhythms, 2018 [ref 7 in ms]. From those studies, we have learned how hepatic and muscle clock gene expression changes in response to time-restricted HFD fed conditions (indicated in Line 81-84). Regarding the comparison of chow and HFD when both food intake and physical activity are time-restricted, these conditions were tested, with same food, same sex, same running wheels and same personnel and have been previously published (see A. Shiba et al., “Synergy between time-restricted feeding and time-restricted running is necessary to shift the muscle clock in male wistar rats,” Neurobiol. Sleep Circadian Rhythms, 2024 [ref 13 in ms]. For the better description of relative importance of these studies, we added “that was performed with the exact same experimental setup and design as the current study” in Line 261-262.
Comments 4: You also have two control groups of animals, one on a standard diet and the other on a HFD, and that's fine, but you should emphasize that the feeding of these animals was ad libidum and not time restricted, because you were only monitoring standard conditions and the shift in the circadian clock by simply changing the type of diet. Response 4: Indeed, we changed the wording in the relative Materials and Methods section accordingly to highlight the ad-libitum condition: Line 136: with ad-libitum food access.
Comments 5: The last sentence of the abstract would be an excellent conclusion if you had LRLF and DRDF on a standard diet in this trial or if you had LF-H and DF-H. You cannot draw this conclusion based on the description of the groups in the second paragraph of the abstract alone. Response 5: The reviewer is correct that we cannot make this statement solely based on the results reported in the current manuscript. Therefore, we changed this sentence accordingly: “Together with (our) previous results, these data suggest that” eating healthy, but being active at the wrong time of the day, can be as detrimental as eating unhealthy and being active at the right time of the day.
Comments 6: Explain Zeitgeber time and acrophase to the readers, i.e. connect ZT0 with a point in standard time (morning point). The topic is interesting and will surely be read by those who are not familiar with all the technical terms. Response 6: Thank you for the suggestion. We edited the text accordingly: Line 109-110: “(i.e. lights on, and start of innate inactive phase)” after ZT0, and “(i.e. lights off, and start of innate active phase)” after ZT12. Line 212: “(i.e. the time point corresponding to the peak of the fitted gene oscillation curve)” after acrophase.
Comments 7: Explain why the animals in the NR and NR-H group were not used for RT-qPCR when they are the control for the standard circadian rhythm, i.e. the shift in it induced only by food. Response 7: Although the NR was indeed not included for the RT-qPCR results, the NR-H group actually was included. We chose to exclude the NR group because we only had a sample size of n=3 per time point for this group. Therefore, we displayed the comparison including an NR group from the previous experiment only in the supplemental figures and decided not to make a direct comparison between NR from the current experiment and the HFD time-restricted groups. In addition, the clock gene expression changes between chow and HFD NR have been reported in references 6 and 7, previous studies from our group.
Comments 8: It is awkward to say body weight development (Fig 1B), body weight evolution (line 272) Response 8: We replaced ”development” with “growth”, and “evolution” was replaced with “increase”. |
Round 2
Reviewer 2 Report
Comments and Suggestions for Authors
Dear authors,
Your corrections contributed to the comprehensibility of the work. However, what is most important for this study is proper controls. You state that you have them in previous works and that you emphasized this in the part of the text that I copied into this answer:
The expression rhythm of Plin5 was shifted in the light phase time-260
restricted feeding and running groups under both chow and HFD conditions (See Fig.5 for an overview 261
of our findings).
I don't see what the sentence in lines 260-261 has to do with the correction that was requested. Indicate that you have controls in the section dealing with methodology and specify that they were done in the papers you cite in the same way as in this paper.
I still think that the authors should state in the Materials and Methods section that some control groups were excluded from this study because they were done in a previous study and that they followed the 3R principle in this paper. This will immediately answer the interested reader why they do not have all control groups in this paper.
Author Response
- Summary
Thank you very much for taking the time to review this manuscript. Please find the detailed responses below and the corresponding revisions/corrections highlighted/in track changes in the re-submitted files. Modifications that were made in response to reviewers comments on the manuscript are highlighted with light green marker.
2. Questions for general evaluation |
Yes |
Can be improved |
Must be improved |
|
Does the introduction provide sufficient background and include all relevant references? |
(x) |
( ) |
( ) |
Is the research design appropriate? |
( ) |
(x) |
( ) |
Are the methods adequately described? |
( ) |
(x) |
( ) |
Are the results clearly presented? |
(x) |
( ) |
( ) |
Are the conclusions supported by the results? |
( ) |
(x) |
( ) |
Are all figures and tables clear and well-presented? |
(x) |
( ) |
( ) |
- Point by point response to comment and suggestions for Authors
Dear authors,
1. Your corrections contributed to the comprehensibility of the work. However, what is most important for this study is proper controls. You state that you have them in previous works and that you emphasized this in the part of the text that I copied into this answer:
The expression rhythm of Plin5 was shifted in the light phase time - 260
restricted feeding and running groups under both chow and HFD conditions (See Fig.5 for an overview 261 of our findings).
I don't see what the sentence in lines 260-261 has to do with the correction that was requested. Indicate that you have controls in the section dealing with methodology and specify that they were done in the papers you cite in the same way as in this paper.
The new wording that we added in the revised manuscript in Lines 261-262 was supposed to read as follows “…that was performed with the exact same experimental setup and design as the current study [13]…”, thus citing our previous paper and stating that the current experiment was performed in the same manner as our previous experiment.
We can only assume that there has been some sort of error of getting line number altered somewhere along the timeline of the process of our new upload until you receiving it. In case that was the case, to indicate the position of the new wording without having to use line numbers, the sentence is added on Page 11, 4th and 5th sentence from the top, below Figure.4.
I still think that the authors should state in the Materials and Methods section that some control groups were excluded from this study because they were done in a previous study and that they followed the 3R principle in this paper. This will immediately answer the interested reader why they do not have all control groups in this paper.
We agree with the reviewer that stating the reasoning of excluding some control groups due to the 3R principle will answer immediately the questions that some readers might have regarding (missing) control groups. We now added the following sentence:
“In order to minimize the number of animals used for the current study according the 3R principle, we did not include no-running (sedentary) controls with time-restricted feeding (LF, DR) and reduced the number of NR animals. Consequence of the reduced number of NR animals was that we could not include them in the current qPCR experiment. However, daily clock gene expression patterns of these three control groups have been reported previously in studies using the exact same experimental setup and design [6], [7], [13].”
in the Materials and Methods section in Lines 139–144.